# Glucocorticoid Receptor Alpha Targets SLC2A4 to Regulate Protein Synthesis and Breakdown in Porcine Skeletal Muscle Cells

**DOI:** 10.3390/biom11050721

**Published:** 2021-05-12

**Authors:** Xiao-Li Du, Wei-Jing Xu, Jia-Li Shi, Kai Guo, Chang-Tong Guo, Rong Zheng, Si-Wen Jiang, Jin Chai

**Affiliations:** College of Animal Science and Technology, Huazhong Agricultural University, Wuhan 430070, China; 15207131682@163.com (X.-L.D.); 2385685069@163.com (W.-J.X.); realscary@foxmail.com (J.-L.S.); 15038027365@163.com (K.G.); Gct792808035@163.com (C.-T.G.); zhengrong@mail.hzau.edu.cn (R.Z.); jiangsiwen@mail.hzau.edu.cn (S.-W.J.)

**Keywords:** pig, glucocorticoid, GRα, protein deposition, SLC2A4

## Abstract

In the presence of stress, the hypothalamic-pituitary-adrenal (HPA) axis activity can be enhanced to promote the secretion of a large amount of glucocorticoids (GCs), which play an important role in the anabolism and catabolism of skeletal muscle. When the endogenous and exogenous glucocorticoids are deficient or excessive, the body will produce stress-related resistance and change the protein metabolism. In this study, we investigated the effect of GC receptor GRα on protein breakdown and synthesis in porcine skeletal muscle cells (PSCs). Overexpression of GRα was shown to increase the expression of protein degradation-related genes, while knockdown of GRα decreased the expression of these genes. Additionally, we found a relationship between GRα and solute carrier family 2 member 4 (SLC2A4), SLC2A4 expression level increases when stress occurs, suggesting that increasing SLC2A4 expression can partially alleviate stress-induced damage, and we found that there is a combination between them via luciferase reporter assays, which still needs to be confirmed in further studies.

## 1. Introduction

Glucocorticoids (GCs) are a type of steroid hormones secreted by the adrenal cortical bundle, and their secretion could be increased under stress [1]. The feedback regulation of GCs on the hyperactivity of the hypothalamic-pituitary-adrenal (HPA) axis is considered to be mediated through two different intracellular receptors: the mineralocorticoid receptor (MR) and GC receptor (GR) [2]. GR, a family member of conserved nuclear receptors, is a nuclear transcription factor widely present in various somatic cells [3] and can only be activated under high concentration GC stimulation [4]. GRα and GRβ are two subtypes of GC. In most cells, GRα is more abundant than GRβ, and it can bind to GC and activate the response of related genes [5,6].

Under normal circumstances, insulin in the liver triggers the rapid uptake and oxidative decomposition of glucose [7], while under stress, GC acts as an antagonizing substance in the anabolic function of insulin to promote glycogen decomposition and gluconeogenesis in the liver [8,9,10]. Increasing the GC content can regulate the metabolism of glucose, fat and protein in the body [11]. Additionally, GC plays an important role in different substance metabolism. In skeletal muscle, GC reduces the protein synthesis rate and increases the protein breakdown rate, both of which can cause skeletal muscle atrophy [12,13]. Meanwhile, animal protein synthesis can be affected under chronic stress. In our previous study, the weight gain rate of cortisol-fed piglets was found to be significantly lower than that of the control group [14]. Stress hormones lead to slower growth in pigs, which is closely related to the rate of protein deposition and breakdown in muscles. Moreover, glucose is an important fuel for contracting muscle, and normal glucose metabolism is vital for muscle health. Glucocorticoids regulate multiple aspects of glucose homeostasis in skeletal muscle. Solute carrier family 2 member 4 (SLC2A4) plays an important part in glucose metabolism, which controls glucose transport into muscle tissues in response to insulin after exercise [15,16]. In our previous experiment on cortisol-fed piglets, we found that the expression of SLC2A4 in the cortisol group was significantly higher than that in the control group via transcriptome sequencing. Therefore, we speculate that stress may affect the decomposition or deposition of protein through GRα regulation of SLC2A4, leading to a slower growth rate.

In this study, we investigated the impact of GRα on protein deposition-related genes in porcine skeletal muscle cells (PSCs) and explored the relationship between GRα and SLC2A4. 

## 2. Results

### 2.1. Effects of Glucocorticoid and Its Antagonist on Protein Deposition in Pig Skeletal Muscle Cells

Porcine skeletal muscle cells (PSCs) were cultured in 6-well plates and treated with four different Dexamethasone (DEX)concentrations (0, 0.1, 1 and 10 μM) for 48 h. The highest expression of GRα was found to occur at 0.1 μM DEX (Appendix A Appendix A). *Atrogin-1*, *FOXO1* and *MSTN* have been reported to affect protein deposition [17]. For further investigation, we used DEX and GC nuclear receptor antagonists (RU486) to treat PSCs. Based on the results of RT-PCR and Western blot, the mRNA expression of these genes increased when treated with DEX, but decreased after adding an equal concentration of RU486. We used the SUnSET non-radioactive method to detect the protein synthesis of the cells after treatment with DEX and its antagonist, and the protein synthesis rate was shown to decrease after adding DEX (Figure 1).

### 2.2. Effect of Glucocorticoid Receptor GRα on Protein Deposition

Previous studies have shown that DEX affects the expression of protein deposition-related genes in PSCs, so we further studied the effects of GRα on protein deposition-related genes in PSCs by the overexpression and interference of GRα. The results of overexpression and interference of GRα are shown in Figure 2. After overexpression of GRα, the protein concentration was shown to decrease significantly in PSCs (Figure 3A), coupled with an increase in the expression of the inhibitory protein deposition-related gene and a decrease in the protein synthesis rate (Figure 4). After the intervention of GRα, the protein concentration increased (Figure 3B), coupled with a decrease in the expression level of the inhibitory protein deposition-related genes and an increase in the protein synthesis rate (Figure 5). These results indicate that when stress occurs, both the level of glucocorticoids and the corresponding expression levels of their receptors are elevated, resulting in a decrease of protein deposition in the muscle.

### 2.3. Effect of GRα on SLC2A4 Expression and Activity Analysis of SLC2A4 Promoter

The differentially expressed genes were screened according to the sequencing results of the previous transcriptome analysis. The mRNA expression levels of *SLC2A4*, *FKBP5* and *GPX2* were markedly increased, in contrast to no significant difference in the expression level of *FHL3* and *USP18* in the longissimus dorsi muscle of cortisol-fed piglets versus the control group (Appendix A Appendix A). The potential effect of GRα on differentially expressed genes was explored by overexpression and interference of GRα in porcine skeletal muscle cells. Overexpression of GRα significantly increased the mRNA and protein level of the SLC2A4 gene, while interference of GRα significantly decreased its mRNA and protein level (Figure 6). This indicates a certain relationship between GRα and *SLC2A4*, i.e., the increase in the level of glucocorticoids in the body can promote the expression of SLC2A4, thereby regulating the response in the body. 

The active site in the *SLC2A4* promoter was predicted using the Neural Network Promoter Prediction (NNPP) online software, and the binding site of the transcription factor GRα and *SLC2A4* was predicted with the 1167 bp sequence containing the promoter active site using the JASPAR online software (Appendix A). Four deletion fragments upstream of the translation initiation site were amplified based on the predicted binding site, which were 1167 bp, 827 bp, 487 bp, and 294 bp, and each was inserted into the pGL3-Basic carrier, named S1, S2, S3 and S4, respectively. The activity of the deleted fragments was determined by transfecting them into pig embryonic kidney cells (PK) and PSC. The luciferase reporter system was used to detect the dual-luciferase activity of these fragments. The S3 fragment was detected to be most active (Figure 7A,B), so the S3 fragment and the GRα overexpression vector were co-transfected for further analysis. Compared with the control group, the co-transfection group showed a significant increase in activity (Figure 7C,D), indicating that the S3 fragment has a site binding to GRα. To determine whether GRα directly binds to *SLC2A4*, we verified a GR binding site on the S3 fragment through site-directed mutagenesis (Figure 7E). Meanwhile, we found that the mutation of the binding site did not completely reduce the activity of the luciferase reporter, indicating that there might be other GRα binding sites or other modes of action on the S3 fragment.

### 2.4. Effect of SLC2A4 on Protein Deposition

In order to study the role of SLC2A4 in protein deposition, an overexpression vector of *SLC2A4* was constructed, and the expression levels of *Atrogin-1*, *MSTN*, *FOXO1* and *IGF-1* were detected after overexpression of *SLC2A4* in PSC. The expression level of *Atrogin-1* was shown to decrease significantly, in contrast to an increase in the expression of *FOXO1* and *MSTN* (Figure 8A–E). The protein expression level of Atrogin-1 was detected to decrease significantly (Figure 8F–H). Under low glucose and hypoxia conditions, the AMPK signaling pathway was activated, thereby inhibiting the mTOR signaling pathway. Overexpression of *SLC2A4* and analysis of the genes in the mTOR signaling pathway revealed an increase in the phosphorylation level of mTOR and its downstream genes S6K1 and 4EBP1, indicating that the increased expression of the glucose transporter *SLC2A4* can alleviate the decrease of protein deposition caused by stress (Figure 8I).

## 3. Discussion

There are a variety of stresses in the growth and development of livestock. The occurrence of stress will induce a series of reactions in the body, such as the increase in the level of free radicals in the cells, and the body’s antioxidant system such as SOD cannot completely remove these free radicals, causing cell apoptosis, tissue damage, increased muscle fiber gap, and decreased meat quality [14,18]. In addition, when stress occurs, the body’s hormone secretion is abnormal, thus reducing the digestion and absorption capacity of nutrients, which in turn affects the nutrient utilization rate and growth performance of the body [19]. The neuroendocrine system plays an important role in maintaining homeostasis, and glucocorticoids and insulin perform important functions in maintaining energy metabolism in the body [20]. Under external environmental stresses, such as pre-slaughter stress, transport stress, etc., the HPA axis can increase the level of glucocorticoids in the blood through interaction with insulin, adrenaline, glucagon and sympathetic nerves, leading to decreased insulin levels, mobilization of fat and muscle tissue, and enhanced hepatic gluconeogenesis [21]. Glucocorticoids can reduce the level of protein synthesis, and many genes are reported to affect protein synthesis, such as *Atrogin-1*, *MSTN*, *FOXO1* and *IGF-1* [17]. Among them, the Atrogin-1 gene has a high level of expression in the event of muscle atrophy. MSTN is an important factor regulating muscle growth and development, and an increase in its expression will cause small muscle mass blocks in animals [22,23]. The FOXO family is involved in the regulation of various cellular functions, such as transcriptional metabolism, apoptosis and cell cycle. In this family, *FOXO1* can regulate cell differentiation. When constructing a specific transgenic mouse overexpressing *FOXO1*, the mutant mouse was found to be lower than the wild-type mouse in body weight and skeletal muscle weight, and higher in protein degradation in skeletal muscle [24,25,26]. In this study, we investigated the effect of glucocorticoids on protein deposition by simulation on PSC, and found that the related genes varied in their expression. Specifically, when the stress state leads to an increase in the glucocorticoid level, the protein synthesis in the body will decrease, and so will protein deposition.

Despite the potential involvement of calcium-dependent protein degradation, the glucocorticoid-stimulated muscle protein breakdown is primarily caused by ubiquitin-proteasome-dependent proteolysis, and under certain catabolic conditions, such as sepsis, the interaction between glucocorticoids and pro-inflammatory cytokines is also important for stimulating muscle protein breakdown. In a previous study on muscle atrophy, the activation of muscle protein degradation was found to require endogenous glucocorticoids. Specifically, when given the physiological doses of glucocorticoids, the adrenalectomized mice showed decreased PI3K activity and progressive muscle atrophy. These reactions are related to an increase in GRα expression. The muscle-specific deletion of GRα in mice showed an increase in the activity of PI3K, and the non-genomic effects of GR are assumed to contribute to the activation of muscle protein degradation [27]. In this study, overexpression of GRα reduced the protein concentration, suggesting the decrease of protein deposition after overexpression of GRα. Analysis of the mRNA and protein expression levels of genes related to protein degradation revealed an increase in their expression after overexpression of GRα, while an increase in their protein concentration and a decrease in their expression after the interference of GRα. This result supported our hypothesis that GR promotes muscle protein degradation and reduces protein deposition.

*SLC2A4* is an insulin-sensitive transporter that can be inserted into the intracellular vesicle membrane and translocated to the plasma membrane under insulin stimulation, thereby increasing the glucose uptake of cells. Muscle contraction can stimulate the translocation of *SLC2A4* in skeletal muscle and induce a large translocation when contraction is tight [28,29]. Under chronic stress, high levels of glucocorticoids in the body can cause high blood pressure. Zheng Xiang et al. detected the blood glucose levels in mice after injection of DEX, and found an increase in the blood glucose levels of the experimental group [30,31]. Meanwhile, we also found that the expression level of *SLC2A4* was increased in the longissimus dorsi muscle samples of cortisol-fed piglets or after overexpression of GRα in PSC. The gene transcription is promoted by the combination of glucocorticoid and glucocorticoid reaction element of the gene promoter region [32]. The expression of *SLC2A4* was increased by overexpression of GRα, but decreased by the interference of GRα, inferring that GRα may also have a relationship with *SLC2A4*. For further analysis, we predicted a binding site for GRα on *SLC2A4*, which was investigated by dual-luciferase reporter and co-transfection with GRα overexpression vector. We verified that there is a GRα binding site on the S3 fragment through site-directed mutagenesis (Figure 7E), and found that the mutation of the binding site could not completely reduce the activity of the luciferase reporter, indicating that there may be other GRα binding sites or other modes of action on the S3 fragment. GRα controls transcription by two major modes of action: (1) binding GRα homodimers to glucocorticoid response elements (GREs) in the regulatory sequences of GRα target genes, and (2) modulating the activity of other transcription factors, such as AP-1, NF-κB and Stat5, independently of direct DNA contact, a process designated as cross-talk, which has been previously confirmed [33] (Reichardt et al., 2016). So we speculated that there might exist a second mode of action between GRα and *SLC2A4*, and it needs to be illustrated by our further experiment. It was initially shown that GRα binds to *SLC2A4* and acts to increase *SLC2A4* expression and glucose transport level to alleviate the effects of glucocorticoid-induced reduction in protein deposition. *SLC2A4* is in the insulin signaling pathway, and skeletal muscle is the main site of insulin-stimulated glucose uptake. Glucose transport is found to increase in response to the contraction in isolated skeletal muscle, indicating that the signal transduction pathway caused by the deficiency of internal cellular energy partially acts on contraction [34]. In our study, overexpression of *SLC2A4* was found to decrease the expression of the proteolytic gene *Atrogin-1*, while increasing the expression of the other two genes (*FOXO1* and *MSTN*), indicating that glucose stress can partially alleviate the effects of protein deposition under stress. We think a feedback mechanism may exist: under stress, the body will reduce protein synthesis, but will perform feedback regulation to make the body as balanced as possible. Buffering provides an alternative regulatory strategy to negative feedback [35,36,37]. Additionally, the mTOR signaling pathway plays an important role in signaling pathways related to protein synthesis. mTOR promotes protein synthesis through phosphorylation of two key effectors, p70S6 kinase (S6K1) and elF4 binding protein (4EBP) [38]. By overexpressing *SLC2A4*, we detected the mRNA expression in the mTOR signaling pathway and then found an increase in the phosphorylation level of mTOR and S6K1 protein. In a previous study on muscle atrophy, mTOR activation significantly decreases the expression level of *Atrogin-1* and efficiently counteracts the catabolic processes provoked by glucocorticoids [39], implying the *SLC2A4* reduces the expression level of Atrogin-1 through the mTORC1 signaling pathway, thus alleviates the decrease in protein deposition caused by stress. 

In summary, the HPA axis of the body is shown to be activated during stress, leading to increased levels of glucocorticoids and their receptors, which increases the expression of genes *Atrogin-1*, *MSTN* and *FOXO1* and promotes protein breakdown, resulting in reduced protein synthesis and inhibition of protein deposition. Meanwhile, elevated levels of glucocorticoids can increase the expression of the glucose transporter gene *SLC2A4*. Overexpressed *SLC2A4* promotes mTOR signal and reduces the expression level of the protein degradation gene *Atrogin-1*, which partially alleviates the decrease of protein deposition in porcine skeletal muscle cells (Figure 9).

## 4. Materials and Methods 

### 4.1. Animals and Samples

The big white piglets of 2–3 days were provided by the Farm of Huazhong Agricultural University (Wuhan, China). After the piglets were slaughtered, the longissimus dorsi and leg muscles were taken to separate for the primary skeletal muscle cells. Finally, the skeletal muscle cells were grown in incubators at 37 °C and 5% CO2, and proliferating cells were cultured in Dulbecco’s Modified Eagle’s Medium (DMEM) supplemented with 10% fetal bovine serum (DMEM, Gibco, Grand Island, NY, USA).

Animals: All experimental animal procedures in this study were performed according to the guidelines of Good Laboratory Practice, and the animals were supplied with nutritional food and sufficient water. Animal feeding and tests were conducted based on the National Research Council Guide for the Care and Use of Laboratory Animals and approved by the Institutional Animal Care and Use Committee at Huazhong Agricultural University.

Isolation and culture of primary skeletal muscle cells: Primary myoblasts were isolated and cultured as described previously [40]. Primary skeletal muscle cells were isolated from the longissimus dorsi skeletal muscles of 2–3 day old piglets, minced and digested in a mixture of type I collagenase and DMEM (DMEM, Gibco, Grand Island, NY, USA).

### 4.2. Total RNA Preparation and cDNA Synthesis

Total RNA was isolated at 48 h after cell transfection, using total RNA extraction kit (Omega bio-tek, Norcross, GA, USA) according to the manufacturer’s protocol. The RNA integrity was checked using denaturing gel electrophoresis and the RNA concentration was measured with a NanoDrop 2000 spectrophotometer (Thermo Scientific, Waltham, MA, USA). Total RNA was reverse transcribed using a RevertAid^TM^ First Strand cDNA Synthesis Kit (Thermo Scientific, Waltham, MA, USA). 

### 4.3. Quantitative Real-Time PCR

The specific fluorescent quantitative PCR primers were designed using cDNA as a template; the mRNA level was quantified using β-actin gene as an internal reference; the real-time quantitative PCR experiments were performed on the CFX384 Touch^TM^ fluorescence quantitative PCR instrument using SYBR Green qPCR Master Mix (Bio-Rad). Finally, the 2^-^^ΔΔCt^ method was used for data analysis, and one-way analysis of variance was performed to determine the significance at *p* < 0.05 (significant) and *p* < 0.01 (extremely significant). Primer sets are listed in Table 1.

### 4.4. BCA Method for Detecting Protein Concentration

The protein concentration was determined using the BCA Protein Concentration Test Kit (Thermo Fisher Scientific, Waltham, MA, USA) according to the procedure provided by the supplier.

### 4.5. Plasmid Construction, siRNA Synthesis, and Cell Transfection

Briefly, For the GRα and *SLC2A4* overexpression plasmids, full-length sequences were cloned into the pCMV-HA plasmid. Meanwhile, various amounts of *SLC2A4* promoter sequences were amplified from pig genomic DNA and cloned into the pGL3-Basic vector to generate luciferase reporter plasmids. Full-length GRα and *SLC2A4* sequences were amplified with full-length-F/R primers (Table 2) and truncated *SLC2A4* sequences were amplified with *SLC2A4*-F/R primers (Table 2). GRα siRNA was synthesized by Shanghai GenePharm Company, using the following GRα siRNA sequences: GRα siRNA (sense): GCUACUCAAGCCCUGGAAUTT; GRα siRNA (antisense): AUUCCAGGGCUUGAUCAGCTT. For cell transfection, primary cells were transfected with 4 μg of expression vectors or 10 μL of siRNA oligo using Lipofectamine 2000 (Invitrogen, Carlsbad, CA, USA) in each well of a 6-well plate.

### 4.6. Western Blot Analysis

Briefly, PSC was washed with PBS and lysed in RIPA lysis buffer (Beyotime Biotechnology Company, Shanghai, Chian). Next, 20 μg of total protein was resolved by 10% sodium dodecyl sulfate-polyacrylamide gel electrophoresis and electro-transferred onto a poly-vinylidene fluoride (PVDF) (Millipore, Burlington, MA, USA) membrane. The PVDF membrane was blocked in 5% skim milk powder dissolved in TBST for 90 min at room temperature. Primary antibodies were applied in sealing fluid at 4 °C overnight. Subsequently, the PVDF membrane was washed with TBST and stained with the appropriate HPR-labeled secondary antibodies (goat anti-rabbit or mouse) for 1 h at room temperature. After washing with TBST, the membrane was analyzed using the ECL Reagent (Beyotime Biotechnology). The antibodies used included: GAPDH (GB11002) antibody purchased from Servicebio, Wuhan, China; β-Actin (AC037), MSTN (A6913), *SLC2A4* (A7637), Atrogin-1 (MAFbx) (A3193), GRα (NR3C1) (A2164) and secondary antibody purchased from Abclonal Technology, Wuhan, Chian; P-S6K1 (AF3228, Affinity Biosciences), P-mTOR (S2448, Cell Signaling Technology, Danvers, MA, USA), and mouse puromycin antibody 12D10 (320591, Millipore).

### 4.7. SUnSET Non-Radioactive Method for Detecting Protein Synthesis Rate

In this study, the surface sensing of translation (SUnSET) non-radioactive method was used to measure the protein synthesis rate of porcine skeletal muscle cells [40]. After transfection with GRα overexpression vector and GRα interference fragment, the cells were treated separately with DEX and RU486 for 48 h, followed by the addition of 1μg/mL puromycin (Beyotime Biotechnology, Shanghai, China) and treatment of 30 min. Finally, the change of puromycin was detected using Western blot. 

### 4.8. Luciferase Reporter Assays

PSC was seeded in 24-well plates until reaching sub-confluence, followed by transient co-transfection with a luciferase reporter plasmid (500 ng/well) containing the conserved binding sequence of GRα for detecting the GRα activity, using the pRL-TK plasmid encoding C luciferase as a control. Firefly and Renilla luciferase activity was measured using the Dual-Luciferase Reporter Assay System (Promega, Madison, WI, USA).

### 4.9. Statistical Analysis

The results are presented as means ± standard deviation (SD). Statistical analysis of the groups was performed using Student’s t-test or one-way ANOVA with the LSD post-hoc test. Statistical significance was set at * *p*< 0.05 and ** *p*< 0.01.

## Figures and Tables

**Figure 1 biomolecules-11-00721-f001:**
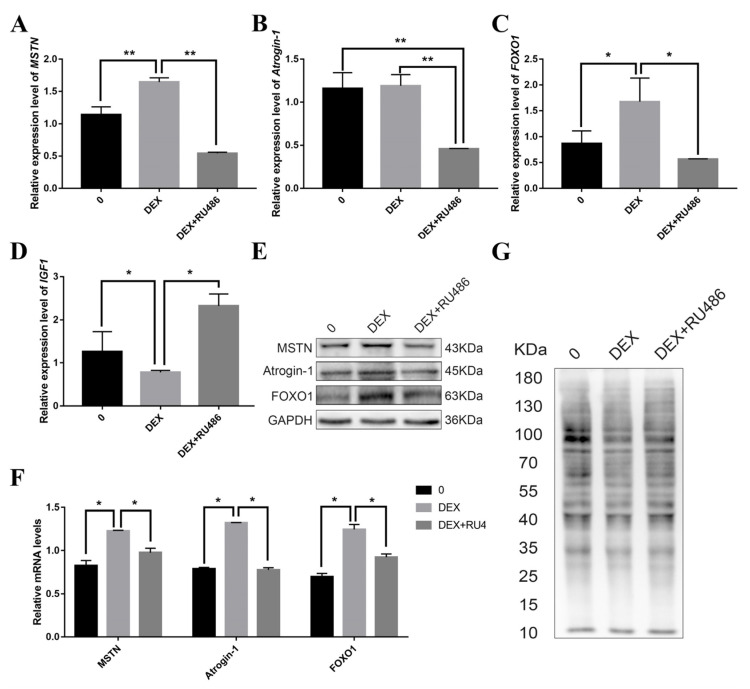
The influence of glucocorticoid and its antagonists on the expression of protein deposition-related genes. (**A**–**D**) The mRNA expression levels of *MSTN*, *Atrogin-1*, *FOXO1*, *IGF1* treated with DEX and RU486; (**E**) The protein expression levels of MSTN, Atrogin-1 and *FOXO1* treated with DEX and RU486; (**F**) Quantification of Western blotting under the treatment of DEX and RU486; (**G**) Effect of glucocorticoid and its antagonist on the protein synthesis rate. *, ** indicate significant difference at *p* < 0.05 and *p* < 0.01, respectively.

**Figure 2 biomolecules-11-00721-f002:**
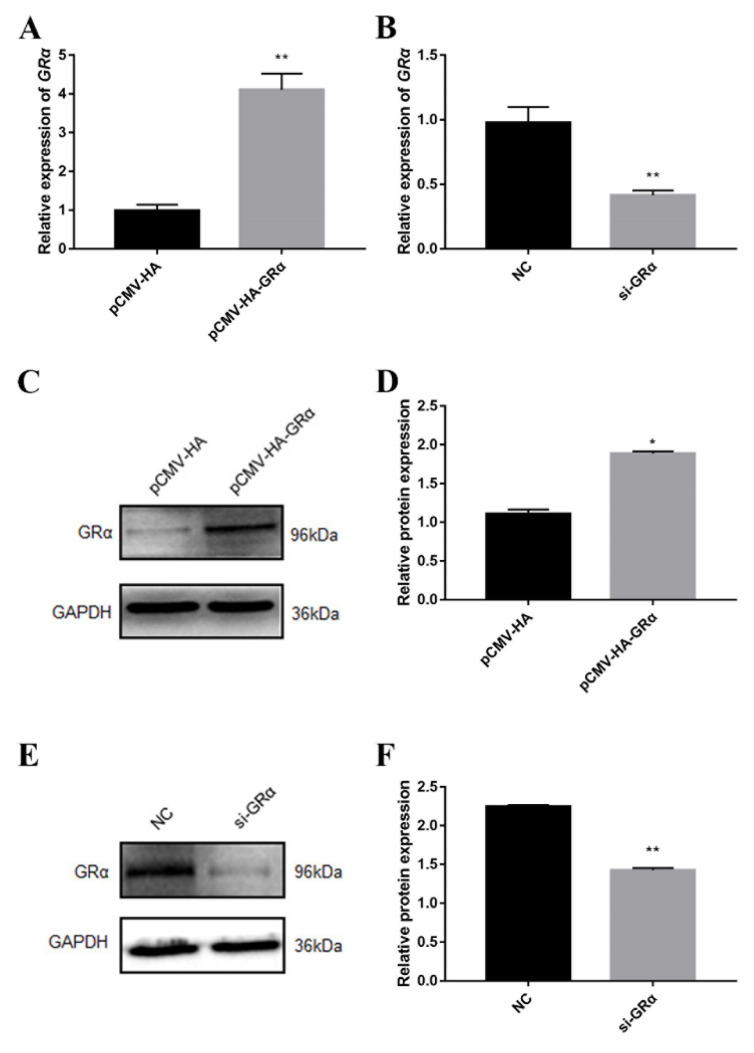
The expression level of GRα *after its* overexpression or interference in porcine skeletal muscle cells (PSCs). (**A**) The mRNA expression level of GRα after overexpression of GRα; (**B**) The mRNA expression level of GRα after interference of GRα; (**C**) The protein expression level of GRα after overexpression of GRα; (**D**) Quantification of Western blotting after overexpression of GRα; (**E**) The protein expression level of GRα after interference of GRα; (**F**) Quantification of Western blotting after interference of GRα. *, ** indicate significant difference at *p* < 0.05 and *p* < 0.01, respectively.

**Figure 3 biomolecules-11-00721-f003:**
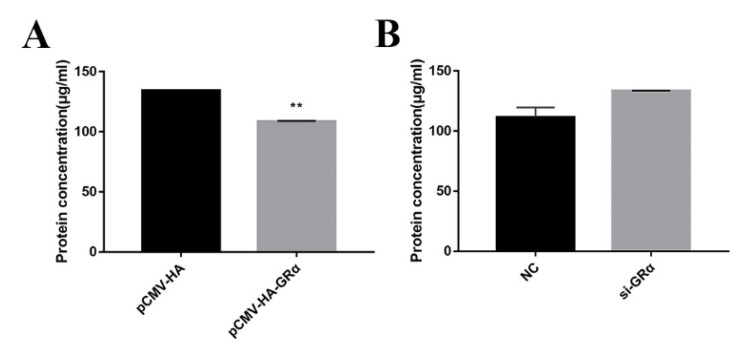
Changes in cellular protein concentration after overexpression or interference of GRα. (**A**) Changes in cell protein concentration after overexpression of GRα; (**B**) Changes in cellular protein concentration after interference of GRα. ** indicate significant difference at *p* < 0.01.

**Figure 4 biomolecules-11-00721-f004:**
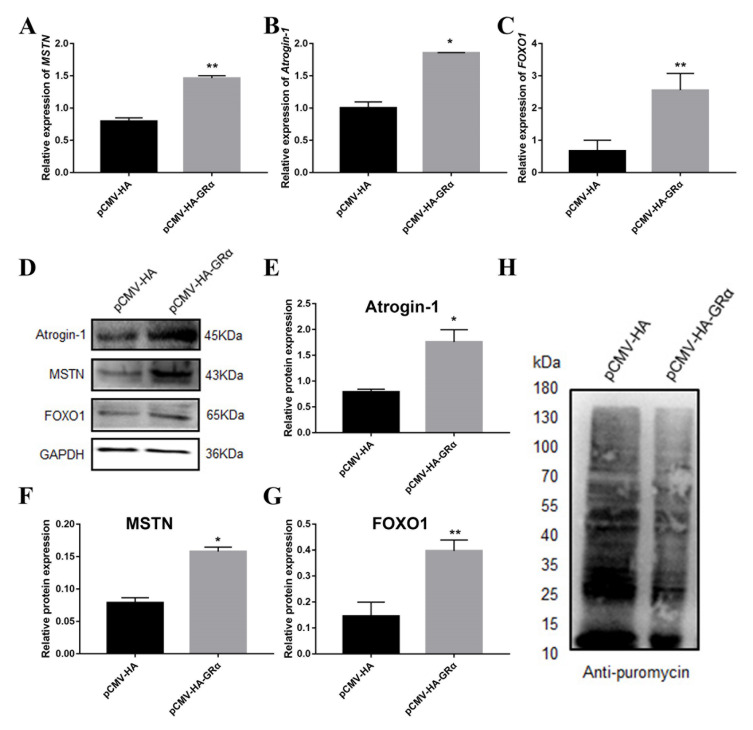
Changes of mRNA and protein levels of protein deposition-related genes. (**A**–**C**) The mRNA expression levels of protein deposition-related genes *MSTN*, *Atrogin-1* and *FOXO1* after overexpression of GRα; (**D**) The protein expression level of protein deposition-related genes MSTN, Atrogin-1 and *FOXO1* after overexpression of GRα; (**E**–**G**) Quantification of Western blotting after overexpression of GRα; (**H**) Effect of GRα overexpression on protein synthesis rate. *, ** indicate significant difference at *p* < 0.05 and *p* < 0.01, respectively.

**Figure 5 biomolecules-11-00721-f005:**
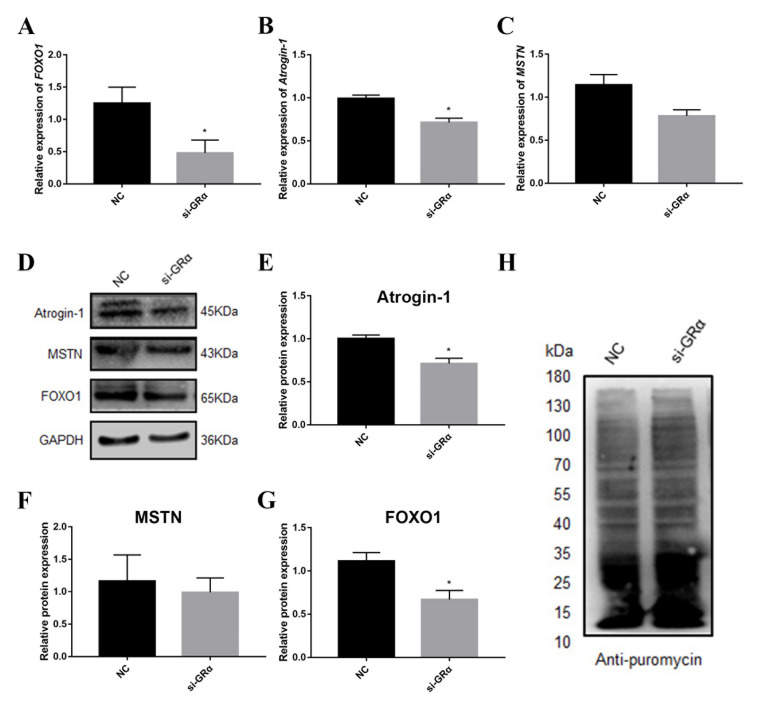
Changes of mRNA and protein levels of protein deposition-related genes after interference of GRα. (**A**–**C**) The mRNA expression level of protein deposition-related genes *FOXO1*, *Atrogin-1* and *MSTN* after interference of GRα; (**D**) The protein expression levels of protein deposition-related genes MSTN, Atrogin-1 and *FOXO1* after interference of GRα; (**E**–**G**) Quantification of Western blotting after interference of GRα. (**H**) Effect of interference of GRα on protein synthesis rate. * indicate significant difference at *p* < 0.05.

**Figure 6 biomolecules-11-00721-f006:**
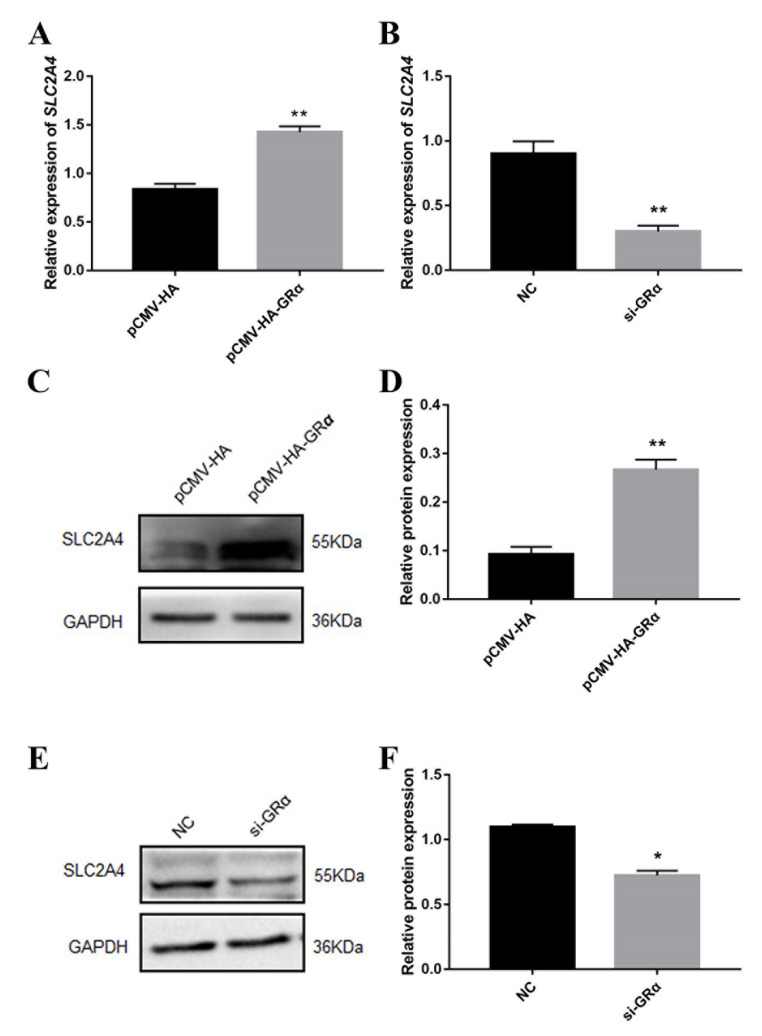
Changes in *SLC2A4* protein level after overexpression or interference of GRα. (**A**) The mRNA expression level of *SLC2A4* after overexpression of GRα; (**B**) The mRNA expression level of *SLC2A4* after interference of GRα; **C**: The protein expression level of *SLC2A4* after overexpression of GRα; (**D**) Quantification of Western blotting after overexpression of GRα; (**E**) The protein expression level of *SLC2A4* after interference of GRα; (**F**) Quantification of Western blotting after interference of GRα. *, ** indicate significant difference at *p* < 0.05 and *p* < 0.01, respectively.

**Figure 7 biomolecules-11-00721-f007:**
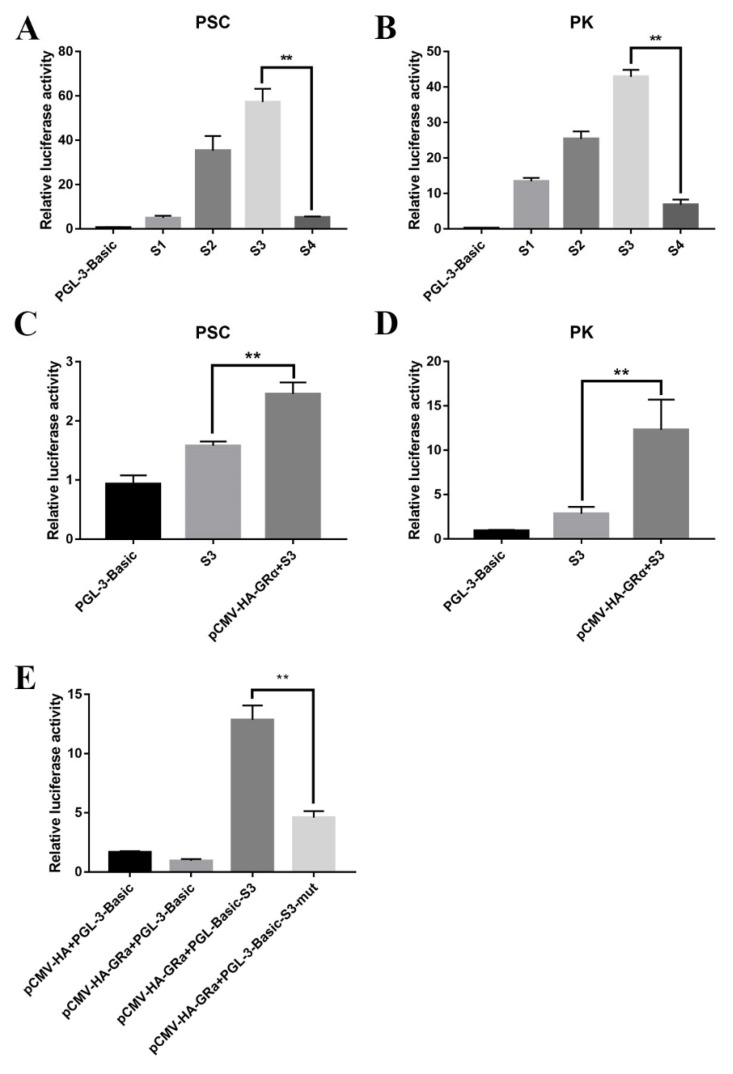
Activity analysis of different deletion fragments of *SLC2A4*. (**A**) The relative luciferase activity of the *SLC2A4* deletion fragment vector in porcine skeletal muscle cells; (**B**) The relative luciferase activity of the deletion fragment of *SLC2A4* promoter in pig embryonic kidney cells (PK); (**C**) The relative luciferase activity after co-transfection of S3 recombinant vector and pCMV-HA-GRα into PSC; (**D**) The relative luciferase activity after co-transfection of S3 recombinant vector and pCMV-HA-GRα into PK. (**E**) Luciferase assays were conducted into PK cells co-transfected with GRα and PGL-3-Basic reporter containing either wild-type (500 ng) or mutant *SLC2A4* binding site (Mut) (500 ng). ** indicate significant difference at *p* < 0.01.

**Figure 8 biomolecules-11-00721-f008:**
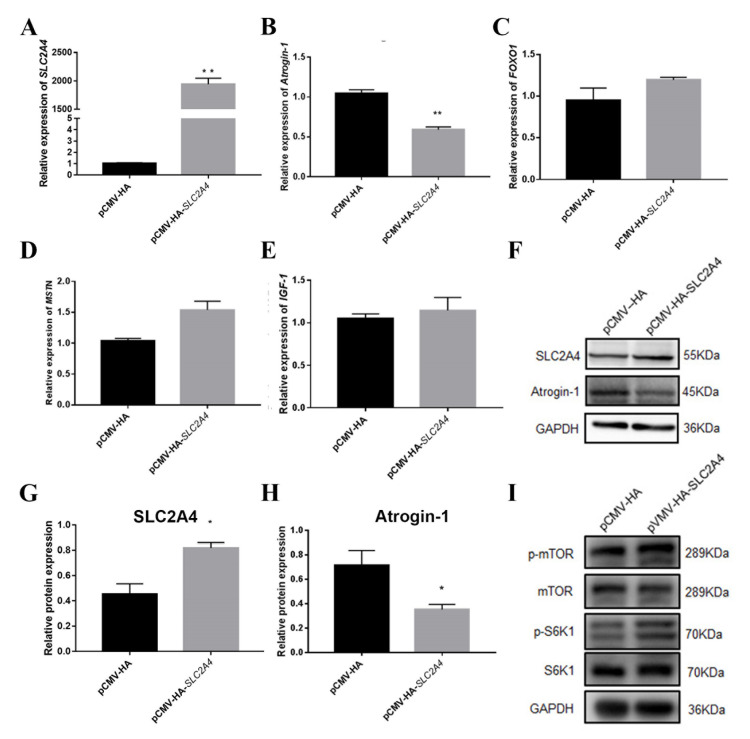
The expression of genes involved in protein deposition and mTOR signaling pathway after overexpression of *SLC2A4*. (**A**–**E**) The mRNA expression levels of *SLC2A4*, *MSTN*, *Atrogin-1*, *FOXO1* and *IGF1* after overexpression of *SLC2A4*; (**F**) The protein expression level of *SLC2A4* and Atrogin-1 after overexpression of GRα; (**G**,**H**) Quantification of Western blotting after overexpression of *SLC2A4*; (**I**) The phosphorylation level of mTOR and S6K1 in mTOR signaling pathway after overexpression of *SLC2A4*. *, ** indicate significant difference at *p* < 0.05 and *p* < 0.01, respectively.

**Figure 9 biomolecules-11-00721-f009:**
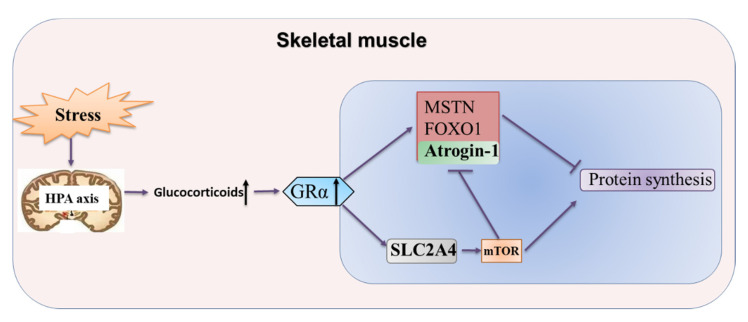
The schematic diagram of glucocorticoid regulating protein deposition process.

**Table 1 biomolecules-11-00721-t001:** Primer sequences used for qPCR.

Gene	Sequence of Primer(5′-3′)	Amplicon Size (bp)
GRα	F:GCTTGCGGTGGACTTTC	126
	R:AGGTCACTTCCCATCACTTTA	
*MSTN*	F:TACCCTCACACTCATCTTGTGC	160
	R:ACCCACAGCGATCTACTACCA	
*Atrogin-1*	F:CAAAGGCTAAGTGATGGCCG	205
	R:GAGGGTAGCATCGCACAAGT	
*FOXO1*	F:AATCGAGTTACGGAGGCATGG	165
	R:TAGGGCCCATCAGCACATTC	
*SLC2A4*	F:TCTCTGTGGGTGGCATGTTC	181
	R:TAGGCACCAATGAGGAACCG	
*FKBP5*	F:GATGGAGTACGGCTTGTCAG	166
	R:CAAGCCCTTCTCATTGGCAC	
*FHL3*	F:CTGTGCAAAATGCAGCGAGT	167
	R:GGAAGTGGCGATCCTCGTAA	
*USP18*	F:TACCTCACCGTCTGGAACCT	195
	R:CAGGGGCTTTGAGTCCATGT	
*IGF1*	F:CTCTCCTTCACCAGCTCTGC	200
	R:TCCAGCCTCCTCAGATCACA	
*β-actin*	F:CCAGGTCATCACCATCGG	158
	R:CCGTGTTGGCGTAGAGGT	

**Table 2 biomolecules-11-00721-t002:** Primers used for plasmid construction.

Gene	Sequence of Primer(5′-3′)
GRα-full length sequence	F: ATGGACCCCAAGGAATCGCTGACCC
	R: TCACTTTTGATGAAACAGAAGTTTT
*SLC2A4*- full length sequence	F: GGCTACACCTGTGGCATATG
	R: CTTGTCTTAGGAGCTGGAGG
*SLC2A4*-S1-1167bp fragment	F: CGGGGTACCGAGGCCCGTTTTCCCAGCCG
	R: CCGCTCGAGCTTGTCTTAGGAGCTGGAGG
*SLC2A4*-S2-827bp fragment	F: CGGGGTACCCTAGGAACGGAATTTCCTGT
	R: CCGCTCGAGCTTGTCTTAGGAGCTGGAGG
*SLC2A4*-S3-487bp fragment	F: CGGGGTACCGTGGGCGGAGTCTTCGCACT
	R: CCGCTCGAGCTTGTCTTAGGAGCTGGAGG
*SLC2A4*-S4-294bp fragment	F: CGGGGTACCCTTCTGGGGTGTGCGGGCT
	R: CCGCTCGAGCTTGTCTTAGGAGCTGGAGG
*SLC2A4*-S3-mut fragment	F: TCGCCCCTACTGACTTCTGCCCGCCAGGCT
	R: GGGCAGAAGTCAGTAGGGGCGACGGGGG

## Data Availability

The data presented in this study are available in the article and Appendix A.

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
