# Peer review of "Glucocorticoid Receptor Alpha Targets SLC2A4 to Regulate Protein Synthesis and Breakdown in Porcine Skeletal Muscle Cells"

_biomolecules, 2021, doi:10.3390/biom11050721_

Round 1

Reviewer 1 Report

This study on the effect of the GC receptor GRα on protein breakdown and synthesis in porcine skeletal muscle cells presents interesting and relevant findings. Several changes and clarifications are needed:

  • Please provide background information regarding SLC2A4 in the introduction to highlight the need for the study.
  • The "Institutional Review Board Statement" has not been completed. Please provide information on the ethical permission for the study.
  • The Student's t-test is appropriate to compare two groups. Please use another test in the instances, in which you compare more than two groups. Moreover, please clarify whether the data were normally distributed because this is a requirement for the use of Student's t-test. 
  • More information is necessary in several instances in the methods: the method for primary skeletal muscle cells isolation; the primer sequences; and the methods for overexpression and interference with GRα.
  • I suggest providing more information regarding how the changes in gene expression correlate with functional changes in the cell status. If possible, it would be informative to include results on the functional status of cells. If this is not feasible, please comment on these aspects in the discussion.

Author Response

Sir:
Thank you for your kind comments for our manuscript to Biomolecules (biomolecules-1091176). We appreciate your valuable comments and suggestions to improve it. With regard to your comments and suggestions, please see the attachment.

Reviewer 2 Report

This manuscript describes a study examining the effect of GRalpha on protein synthesis in porcine muscle cells. The effects of GR overexpression and knockdown are shown on three genes Atrogin-1, MSTN, and FOXO1, and SLC2A4, and the authors conclude that SLC2A4 protects against GR-mediated changes in protein synthesis. The explanation of the methods needs more information, some controls should be added, and description of the results needs to be carefully checked for accuracy.

SPECIFIC COMMENTS

For the introduction, I think it would help the reader if the authors explained more about the findings that lead up to the current paper. The end of the introduction briefly mentions a cortisol treatment experiment (Wan 2016) but no details are given (exogenous treatment? species?), and then they mention that SLC2A4 is important but don't mention why. Both of these points would benefit from much more explanation, as they appear to lay the foundation for (and motivate) the current study. In particular, what is the function of SLC2A4 in skeletal muscle? As of now, it is unclear whether the GR effect on protein-deposition genes and the GR effect on SLC2A4 may not be at all related.

In general, many of the methods are unclear or unexplained. Some particular questions:

  1. How were cells dissociated, dead cells removed, cells counted, how many were plated, and how long were they cultured for? Was culture done in steroid-free medium?

  1. Were GR overexpression and GR knockdown experiments done in the presence or absence of Dex? If done without Dex, why is a GR effect expected in the absence of ligand? If done with Dex, please show data from the untreated controls.

  1. As shown in the paper describing the SUnSET method, it would be nice to show no-puromycin control samples to demonstrate the specificity of the western blot data.

  1. Was any animal care protocol followed for these experiments?

  1. What is the GAPDH antibody? I think P04406 is the protein sequence, not an antibody clone, and I'm couldn't find a Google Biotechnology antibody source.

Some of the results seem contradictory, for example in Fig 1 the data in this figure directly contrasts the text. In panel B it is clear that Dex does not increase Atroglin expression. Panel E indicates that MSTN decreases rather than increases with Dex treatment - how was the opposite result obtained for the graph in panel F? Also, were all these data obtained from wild-type or transfected cells?

The SLC2A4 experiments are the most interesting, but this is also confusing. What do the authors mean when they refer to the 'active site' of the promoter? This is distinct from the GR DNA binding consensus sequence? What do the authors mean by 'deletion fragments'? These are sequences from the SLC2A4 promoter that are cloned into the reporter construct? And what does is this reporter construct activated by? As of right now very little is explained for the reader.

Again, for Figure 7, were steroids used when GR and S4 were cotransfected? And was this S3 or S4? The text states that S3 was used but the figure was labeled S4. Addition of Dex and inhibition with RU486 would greatly strengthen these results. Also, ChIP-qPCR is probably important if the authors wish to state that the S3/S4 sequence is bound by GR, otherwise there are other mechanisms that could explain these results that have not been ruled out. Alternatively, the GR binding sequence could be demonstrated by site-directed mutagenesis.

Figure 8 seems to show that SLC2A4 does not mediate the GR-induced changes on the 3 genes of interest (except perhaps MSTN), thus this doesn't seem to be a mediator the protein deposition phenotype. More importantly, does SLC2A4 overexpression or knockdown affect protein synthesis? This seems like the logical final conclusion of this paper, and is more important than whether those three genes are affected. This would show that glucocorticoid-mediated changes in protein synthesis are due to regulation of SLC2A4 expression, and that would be a very useful description. Finally, the conclusion that "SLC2A4 can alleviate the decrease of protein deposition" is in direct contrast to the observation that increased GR causes both elevated SLC2A4 expression and decreased protein synthesis simultaneously.

Author Response

Sir:
Thank you for your kind comments for our manuscript to Biomolecules (biomolecules-1091176). We appreciate your valuable comments and suggestions to improve it. With regard to your comments and suggestions, Please see the attachment.

Reviewer 3 Report

Line 12: glucocorticoid (GC) probably should be glucocorticoids (GCs)

Line 36: “Under normal circumstances, insulin in the liver not only triggers the rapid uptake and oxidative decomposition of glucose, while under stress, GC acts as an antagonizing substance in the anabolic function of insulin to inhibit glycogen decomposition and gluconeogenesis in the liver” This sentence is really confusing; GCs promote gluconeogenesis and increase glycogen storage in liver but the sentence seems to suggest the opposite concept

Line 51: “We found the fluorescence active sites between them” This sentence is not clear

Line 57: Cells are not inoculated but treated

Line 63: “….the protein synthesis rate was shown to decrease after adding DEX (Fig. 1)” but in panel Fig 1G it is impossible to appreciate differences

Line 74: Fig 2 is not related to GRα effects on protein deposition related genes but to  expression levels of GRα after overexpression and interference

Line 79: “After intervention of GRα” probably means after silencing

Line 122: “NNPP software” should be detailed

Line 181: HAP axis should be HPA axis

Line 184: “The expression level of the Atrogin-1 gene in muscle atrophy events is highly expressed in skeletal muscle before atrophy occurs” the sentence should be rewritten clearly

Line 200-202: “but under certain catabolic conditions, such as sepsis, glucocorticoids and pro-inflammatory cytokines, the interaction is also important for stimulating muscle protein breakdown”. the sentence should be rewritten clearly

Line 224: “Meanwhile, we also found that the expression level of SLC2A4 was increased in the longissimus dorsi muscle samples fed with cortisol or after the overexpression of GRα in PSC..” the sentence should be rewritten clearly

Line 225 “samples fed with cortisol” is not correct

Line 297: The SUnSET non-radioactive method for detecting protein synthesis rate should be detailed. No references are included in the paper

Author Response

(The authors gave the same response as above.)

Round 2

Reviewer 1 Report

Dear Authors,

Thank you for the revised version of the manuscript and for addressing the raised concerns. I believe that you have appropriately addressed almost all of the raised points. I have one additional comment: In the statistical analysis subsection, please indicate which post-hoc test you used in conjunction with ANOVA.

Author Response

Sir:

Thank you for your kind comments for our manuscript to Biomolecules (biomolecules-1091176). We appreciate your valuable comments and suggestions to improve it. With regard to your comments and suggestions, we wish to reply as follows:

Point 1: In the statistical analysis subsection, please indicate which post-hoc test you used in conjunction with ANOVA.

Response 1: Thanks for your suggestion. We have added this information in the section of MATERIAL AND METHODS. And highlight the modifications in red.The results are presented as means ± standard deviation (SD). Statistical analyses of the groups were performed using Student’s t-test or 1-way ANOVA with the LSD post hoc test. Statistical significance was set at * p< 0.05 and ** p< 0.01.